# Enhancing Resistant Starch Content of High Amylose Rice Starch through Heat–Moisture Treatment for Industrial Application

**DOI:** 10.3390/molecules27196375

**Published:** 2022-09-27

**Authors:** Chang-Seon Lee, Hyun-Jung Chung

**Affiliations:** Division of Food and Nutrition, Chonnam National University, Gwangju 61186, Korea

**Keywords:** resistant starch, heat–moisture treatment, high amylose rice starch, response surface methodology, physicochemical properties, granular morphology

## Abstract

The objective of the present study is to enhance the resistant starch (RS) content of high amylose rice starch with heat–moisture treatment (HMT) for industrial application. The optimized HMT condition for achieving the highest RS content established using response surface methodology (RSM) was a temperature of 100 °C, moisture content of 24.2%, and a time of 11.5 h. Upon HMT, the RS content increased from 32.1% for native starch to 46.4% in HMT starch with optimized condition. HMT of the starches reduced the solubility and swelling power. The surface of HMT starch granules was more irregular than native starch. The X-ray diffraction (XRD) peak intensity at 2θ = 5° was greatly reduced by HMT, and the peaks at 22.7° and 24.2° were merged. HMT increased the gelatinization temperature and reduced the gelatinization enthalpy. HMT provides a method for the production of high-yield RS2 with high amylose rice starch in industrial application.

## 1. Introduction

With the interest in healthy eating, consumers are focused on increasing intake of plant-based food products and demanding foods which are natural, organic, less processed, and free from additives. Ingredient manufacturers and processed food companies are striving to meet these trends by offering clean label foods to the market [1]. The clean label trend was launched in the United Kingdom in the 1990s. As cholera, foot-and-mouth disease, avian influenza, and other social problems relating to food safety have emerged, consumers are starting to take an interest in labeling that allows them to know exactly what is in their food and its origin. The clean label refers to what is seen as minimally processed and natural or free from negatively associated ingredients such as allergen-related ingredients or additives. Clean-labeled starch indicates that the preparation is free from chemical modification.

Native starches are usually chemically modified to improve their functionality before use in processed food formulations because of the limited functionality and application of the native form. However, chemical modification has become less favorable among consumers because by-products of chemical reagents are present in the modified starch. By contrast, the physical treatments of starch have been gaining increased acceptance. The primary interest in physical modification is related to the clean label starches; they can be classified as natural products even though they are processed [2]. In addition, physical modification in starch has the advantages of its effectiveness, flexibility in relation to heat sources, low cost and the non-generation of chemical residues, which make it an extremely attractive method for industries. heat–moisture treatment (HMT) is a physical technique for starch modification in which starches are treated under restricted moisture content (10–30%) for a certain period of time, from 15 min to 16 h, at a temperature above gelatinization temperature [3,4]. HMT is relatively easy to implement industrial production due to its simple process and ease of management. Whereas, annealing, the hydrothermal treatment of starch in the presence of excess water for an extended period of time, has a high risk of decomposition and multiple heating and cooling cycles, which makes the treatment difficult to industrialize due to complicated processes and high utility costs [3,4,5,6,7].

During HMT of starches, structural changes occur in both the crystalline and amorphous regions. The X-ray diffraction (XRD) pattern of starches with A-type crystallinity is not altered by HMT, while starches with B-type patterns exhibit a change to C-type (A + B) or A-type with an accompanying decrease in the XRD peak intensities [2,5]. HMT induces an increase in the gelatinization temperatures, broadens the phase transition, and decreases the swelling power and amylose leaching from starch granules [2,6]. HMT starches improve heat and shear stability characteristics, like cross-linked starches. These physicochemical changes imply that HMT starch could be used as resistant starch (RS) resources.

Starch in foods can be classified as rapidly digestible starch, slowly digestible starch, and resistant starch (RS), according to its degree and speed of digestion. The RS is a fraction of the starch that is not digested in the small intestine of healthy individuals and enters the colon unchanged, where fermentation by resident microorganisms occurs, resulting in the formation of short-chain fatty acids [7]. RS is classified into subgroups, RS1–RS5. RS1 represents native starch that is in a physically inaccessible with enzymes. RS2 represents potato starch and high amylose starches with B-type structure that are resistant to enzyme digestion. RS3 is retrograded starch during cooling of gelatinized starch. RS4 is chemically modified starch by various types of chemical treatments. RS5 is a type of RS arising from the formation of amylose-lipid complexes [7]. The high-RS foods have gained considerable interest by consumers because its consumption could effectively prevent obesity, diabetes, cardiovascular disease and other chronic diseases. Several studies reported increased RS levels and improved functionality during HMT because RS has physiological characteristics similar to dietary fiber [8,9,10,11]. A lot of insoluble RS4 type starches have been developed as a food source of high-RS foods. High amylose starch can be classified as RS2 due to its high enzyme resistance. As the amylose content increased, the susceptibility to enzymatic hydrolysis was reduced. The high-yield RS prepared by heat–moisture treatment and multiple heating and cooling cycles with high amylose corn starch have been developed and marketed [12,13,14,15]. This process could be suitable for the production of industrial-scale RS2 type starch at low cost.

Most studies on high amylose rice starch have involved the structural and physicochemical properties of rice starch [14,15] and effects of heat–moisture treatment on functional properties of rice starch [4,16]. To date, no studies have explored the suitable method of RS-rich preparation with high amylose rice starch for industrial application. The objective of this study was to establish the optimal conditions for hydrothermal treatment of high amylose rice starch with enhancing RS content in order to improve the industrial applicability of domestic high amylose rice. Furthermore, the physicochemical properties of heat–moisture treated high amylose rice starches were investigated.

## 2. Results and Discussion

### 2.1. Optimization of HMT Conditions for Improving RS Content

A series of experiments were statistically designed using RSM with a Box–Behnken design to establish the stable operating point of the HMT conditions that increased the RS level of high-amylose rice starch. A regression equation was used to describe the correlations between input parameters (temperature, moisture content, and time) and target parameter (RS). Based on the experimental design, three levels of each parameter (temperature: 100, 110, and 120 °C; moisture content: 15, 21, and 27%; reaction time: 8 and 16 h) were studied. The RS values generated under the experimental conditions are shown in Table 1. The RS content of native rice starch was 32.1% and the RS content of the hydrothermal treated starches both increased and decreased between 24.8 to 49.0% (Table 1). Changes in the RS content of starches by HMT are dependent on the starch source and physical treatment condition, such as moisture content, temperature, and duration of treatment [5,6]. Zavareze and Dias [2] reported that higher amylose contents in starch induced higher RS levels during HMT. Huang et al. [4] also found that HMT of high amylose rice starch increased the RS content.

The RS content increased as the moisture content increased under the same temperature and time (Table 1). Previous studies reported that RS content of starch with a higher moisture content during HMT increased compared to the starch with a lower moisture content [17,18], and the RS levels increased with increasing temperature and reaction time [19]. In contrast, the decreased RS on HMT could be attributed to the partial disruption of crystalline structure with evidence of decreased gelatinization enthalpy [6].

The variables and responses had a certain relationship. One variable was fixed to visualize these relationships, and the changes in RS level according to the alteration of the other two variables were described by the contour plots (Figure 1). The contour plots for the effects of treated temperature and moisture level under a constant treatment time are presented in Figure 1A–C. The contour plots showed that an increase in a treatment time significantly increased the RS content. The highest RS level was observed at 16h of treatment (Figure 1C), and the highest RS value (>45%) was shown in the upper-left corner of the figures. In the contour plots for the effects of the treatment temperature and time under constant moisture (Figure 1D–F), the highest RS content was found at 21% moisture content (Figure 1E), and the highest RS value (>44%) was exhibited in the upper-left corner of the figure. The contour plots for the effects of the treatment moisture and time level under a constant temperature indicated that an increase in a temperature resulted in significant decrease in the RS content. The highest RS level was determined at 100 °C (Figure 1G), and the RS value (>45%) was shown in the upper-right corner of the figures. The experimental data was represented as quadratic model by regression analysis of variables. The RS level could be described as a function of temperature (*A*), moisture content (*B*), and time (*C*), as represented by Equation (1):RS (%) = −412 + 4.36*A* + 18.82*B* + 5.25*C* − 0.0107*A*^2^ − 2.215*B*^2^ –0.0548*C*^2^ − 0.0877*AC* − 0.0424*AB* + 0.0239*BC*(1)

The calculated optimal RS value was 46.4% and best potential processing condition was temperature of 100 °C, moisture content of 24.2%, and reaction time of 11 h 33 min. However, the predicted RS level of 46.4% was lower than experimental data. Analysis of variance (ANOVA) was used to evaluate the regression coefficient (R^2^) of the model. The R^2^ value was 0.7929, indicating that the fit between the model and the experimental data was adequate.

### 2.2. Solubility and Swelling Power

The solubility and swelling power of heat–moisture treated starches are presented in Table 1. HMT has been shown to promote changes in the swelling power, solubility, pasting properties, and crystallinity of starch [11,17]. The solubility and swelling power of the starches decreased depending on the HMT conditions. The reduction in these two indices could be attributed to the reinforced molecular chains during HMT and the increased crystalline perfection, which impeded water percolation into the starch [20]. In addition, HMT induced changes in the amorphous regions of starch granule toward increasing rigidity, thereby limiting the starch swelling and solubility [21]. The reduction in swelling power following hydrothermal modification has been attributed to internal rearrangement of the starch granules, which causes further interaction among the starch functional groups [22], resulting in more ordered double helical amylopectin side chain clusters. This accounts for the increased starch crystallinity [23]. The decrease in the solubility of starch by HMT indicated a strengthening of the bonds, with an increase in the interaction among amylose and amylopectin molecules, impeding them from leaching out of the granules [16].

### 2.3. Scanning Electron Microscopy (SEM)

SEM images of native and treated (HMT) rice starch granules are presented in Figure 2. Most of native rice starch granules had a polyhedral shape, and some of the granules showed semicircular and spherical shapes. Most of the high amylose rice starch granules maintained even after HMT, which was in agreement with previous studies [24,25]. However, some starch granules treated at high moisture and temperature showed a marginal crack on the surface and entanglement among starch granules. Similarly, Huang et al. [4] reported that after HMT, the surface of starch granules was partially cracked or rough, and the starch granules were tangled.

Following HMT, the surface of the granules was more irregular, and the granules were more aggregated compared with the native starches. It is assumed that the loss of the physical integrity of the starch granule with distension of the granular surface was caused by a partial gelatinization [16].

### 2.4. XRD

The XRD patterns of native and HMT rice starches are displayed in Figure 3. Native rice starch (NR) showed the typical B-type pattern with distinctly strong peaks at diffraction angles (2θ) of 5.3° and 17.2° and weak peaks at 14.8°, 22.7°, and 24.2°. Cereal starch generally has A-type crystalline pattern, but among cereal starches, high amylose maize and rice starches are mostly reported to have B type pattern [15,26,27]. After HMT, the peak intensity at 5° was greatly reduced, and the peaks at 22.7° and 24.2° were merged into one peak at 23.1°. However, the XRD pattern of the HMT high amylose rice starches retained the prominent peak at 17.2°, suggesting that they maintained the B-type diffraction pattern. Zavareze et al. [16] reported that the diffraction peak intensities of high amylose rice starch decreased after HMT. These results suggest a decrease in the crystalline areas by HMT. This may explain the greater enzymatic susceptibility evidenced by heat–moisture treated starches.

### 2.5. Thermal Analysis

The gelatinization properties of the HMT rice starches were determined by differential scanning calorimetry (DSC). The onset (*T_o_*), peak (*T_p_*), and conclusion (*T_c_*) temperatures of GS was 59.6, 66.5 and 74.1 °C, respectively (Table 2). The gelatinization temperatures of rice starches were increased after HMT depending on moisture content, treatment temperature, and time. Ruiz et al. [25] suggested that the structural changes within the starch granules, including the interaction of amylose–amylose and amylose–lipid, during HMT induced the increase in gelatinization temperature. The gelatinization temperature range (*T_c_**–T_o_*) of native starch was 14.6 °C, which was expanded for heat–moisture treated starches, except for HGS100-21-16. HMT decreased the gelatinization enthalpy (Δ*H*) of high-amylose rice starches. The enthalpy of native starch was 12.5 J/g and those of heat–moisture treated starches were 4.2–9.9 J/g. Previous studies also reported that the gelatinization temperature of heat–moisture treated starch was much higher than that of native starch, and Δ*H* was decreased during HMT [25,28].

## 3. Materials and Methods

### 3.1. Materials

The high amylose rice variety (*Goami*) was provided by Rural Development Administration (Suwon, Korea). The rice starch was isolated from the grain using an alkaline (0.2%) steeping method [29,30]. The amylose content of high amylose rice starch determined by iodine colorimetric method was 32.9%.

### 3.2. HMT Experimental Design

The RSM was used to acquire the highest RS value by optimizing the HMT process, and the Box–Behnken design was used to understand the effects of reaction temperature, moisture content, and time on the RS level of rice starch. The HMT temperature, moisture content, and time were selected as independent variable, and RS level of high amylose rice starch was the dependent variable during HMT. Three levels of each factor were selected from the maximum and minimum values of the preliminary experiment conducted within the generally known HMT conditions. The HMT temperature range is above the gelatinization temperature, and low physical processing temperature are expected to reduce the effect of HMT due to small variations in physicochemical properties, so the temperature of the experiments was set at 100 °C or more. The three different levels for each factor were as follows: temperature, 100, 110 and 120 °C; moisture content, 15, 21, and 27%; time, 0, 8, and 16 h. The design consisted of 14 treatments in which the central point was replicated twice to calculate the experimental error. The results were calculated from nine experimental groups because there was no real data for the 0 h-treated group. A quadratic model with 10 coefficients was used to investigate the response of the RS value by Equation (2):RS (%) = *r*_0_ + *r*_1_*A* + *r*_2_*B* + *r*_3_*C* + *r*_11_*A*^2^ + *r*_22_*B*^2^ + *r*_33_*C*^2^ + *r*_12_*AB* + *r*_13_*AC* + *r*_23_*BC*(2)
where *r*_1_, *r*_2_ and *r*_3_ are the regression coefficients that reflect the nature and extent of the response to the related treatment condition, and *A*, *B*, and *C* are the encoded variables attributed to temperature, moisture content, and time, respectively. The analysis was performed using Minitab software (Version 17.3.1; Minitab Inc., State College, PA, USA).

### 3.3. Preparation of heat–moisture Treated Starches

After measuring the moisture content of raw rice starch, starch samples (20 g) were weighed into the screw cap glass container (100 mL). The starch was adjusted to 15, 21, and 27% moisture content by adding the appropriate amount of distilled water. The samples were mixed thoroughly to breakdown clumps. The containers were sealed, kept at ambient temperature for 24 h to equilibrate the moisture in starch, then placed in a forced air oven (VS-1202D3, Vision Scientific Co., Bucheon, Korea) at 100, 110, and 120 °C. The samples were thermally treated for 8 and 16 h at each temperature. Afterwards the containers were opened, and the starch samples were air-dried until moisture content reached around 10%. The samples were ground and passed through a 120 mesh (<125 mm) sieve, and stored in a desiccator until they were used.

### 3.4. Determination of RS Content

The RS content was determined using the method of Englyst et al. [31] with minor modifications. Porcine pancreatin α-amylase (0.45 g, P-7545, Sigma, St. Louis, MO, USA) was weighed in a beaker and 4 mL of distilled water was added. The suspension was mixed for 5 min and centrifuged at 1500× *g* for 10 min. The supernatant (2.7 mL) was transferred to a beaker and mixed with 0.3 mL of diluted amyloglucosidase (0.24 mL of enzyme and 0.6 mL of distilled water, A-9913, Sigma) and 2 mg of invertase (I-4504, Sigma). This enzyme solution was freshly prepared for each digestion. Starch (100 mg) and 14 glass beads (diameter, 5 mm) were added into glass test tubes. The HCl/guar solution (2 mL) was added to each tube and mixed immediately on a vortex mixer. Then, 4 mL of 0.5 M sodium acetate buffer (pH 5.2) and 1 mL of enzyme solution was added to each test tube and incubated in a shaking water bath (37 °C, 100 rpm). Aliquots (100 μL) were taken and mixed with 1.5 mL of 50% ethanol. To measure the hydrolyzed glucose content, the Megazyme GOPOD assay kit (D-Glucose Assay Kit, Megazyme International) was used. RS was determined as the undigested fraction after 120 min of hydrolysis.

### 3.5. Swelling Power and Solubility

Swelling power and solubility were determined at 80 °C by a modified method of Schoch [32]. Starch (0.25 g, db) was weighed directly into a 50 mL screw cap test tube, and 20 mL distilled water was added. The capped tubes were incubated at 80 °C in a water bath for 30 min. The tubes were cooled in an ice bath and centrifuged at 1000× *g* for 30 min. The supernatant liquid was removed, and starch sediment in the tube was weighed. The supernatant was dried to constant weight in a drying oven (VS-1202D3, Vision Scientific Co.) at 105 °C. The swelling power and solubility were calculated by the equations described by Schoch [32]. The swelling power and solubility were calculated as following equations:(3)$$Swelling\;power\;\left(g/g\right)=\frac{Weight\;of\;wet\;residues\;\left(g\right)}{Dry\;weight\;of\;starch\;\left(g\right)\times \left(100-solubility\right)}$$
(4)$$Solubility\;\left(\%\right)=\frac{Weight\;of\;dried\;supernatnats\;\left(g\right)}{Dry\;weight\;of\;starch\;\left(g\right)}\times 100$$

### 3.6. SEM

The morphology of starches was characterized using a scanning electron microscope (GeminiSEM 500, ZEISS, Oberkochen, Germany). The accelerating voltage was 15 kV for 85 s, and the magnification was 2000×. The starch samples dried in a desiccator were placed on an SEM stub with double-sided cellophane tape and coated with gold/platinum to increase the conductivity.

### 3.7. XRD Pattern

A high-resolution three-dimensional X-ray diffractometer (PANalytical Empyrean, Almelo, The Netherlands) was used to investigate the crystalline structure of the native and HMT starches. The diffractometer was operated at a 40 kV target voltage and a 34 mA target current with a scanning range of 5–40° (2θ) and a scanning rate of 2.0°/min.

### 3.8. Thermal Analysis

The thermal properties of native and HMT starches were determined using a differential scanning calorimeter (DSC 204F1, Netzsch Instruments, Inc., Selb, Germany). A starch sample (6.0 mg, dry basis) and 14 mg of distilled water were loaded into aluminum pan and hermetically sealed. The pans were kept overnight at room temperature. For the analysis, the pans were heated from 5 to 180 °C at a heating rate of 10 °C/min. An empty pan was used as a reference. The onset (*T_o_*), peak (*T_p_*), conclusion (*T_c_*) gelatinization temperatures, and gelatinization enthalpy (ΔH) were determined from the DSC thermograms.

### 3.9. Statistical Analysis

All data were reported as mean values and standard deviations. Differences between factors were evaluated using one-way analysis of variance (ANOVA) and statistical analyses were carried out with Duncan’s multiple range test (*p* < 0.05) using SPSS Statistics (Version 12.0K; SPSS Inc., Chicago, IL, USA).

## 4. Conclusions

To improve the industrial applicability of high amylose rice starch, RS content was increased through HMT and the physicochemical properties of HMT starches were investigated. This study clearly showed that optimal HMT condition for enhancing RS content was established using RSM. HMT leads to an increase in crystalline perfection due to strengthening the bonds between amylose molecules and between amylose and amylopectin of starch granules because the solubility and swelling power of the starch granules decreased, and the gelatinization temperature increased during HMT. Our results could provide useful information for establishing the HMT conditions of high amylose rice starch for the production of high-RS food in industrial application of rice materials.

## Figures and Tables

**Figure 1 molecules-27-06375-f001:**
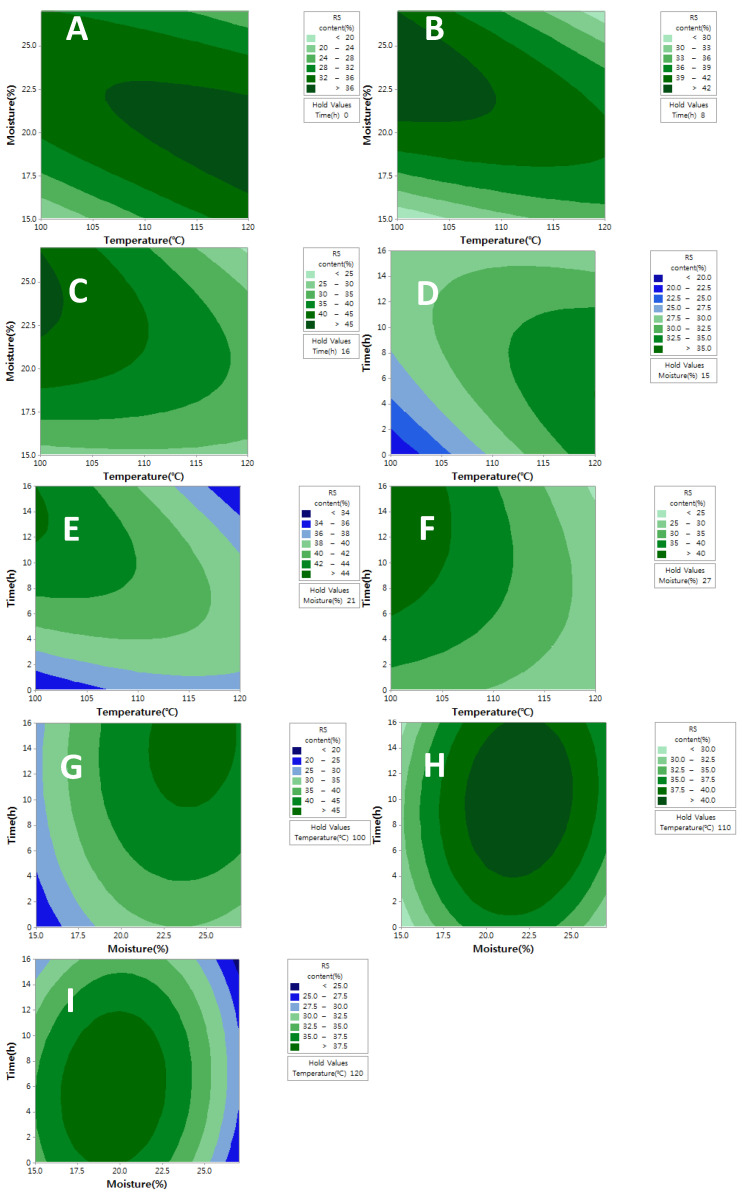
Contour plots for the effects of two variables under a constant one variable. (**A**–**C**): Treatment time was fixed for 0, 8 and 16 h, respectively. (**D**–**F**): Moisture level was fixed at 15, 21 and 27%, respectively. (**G**–**I**): Temperature was fixed at 100, 110 and 120 °C, respectively.

**Figure 2 molecules-27-06375-f002:**
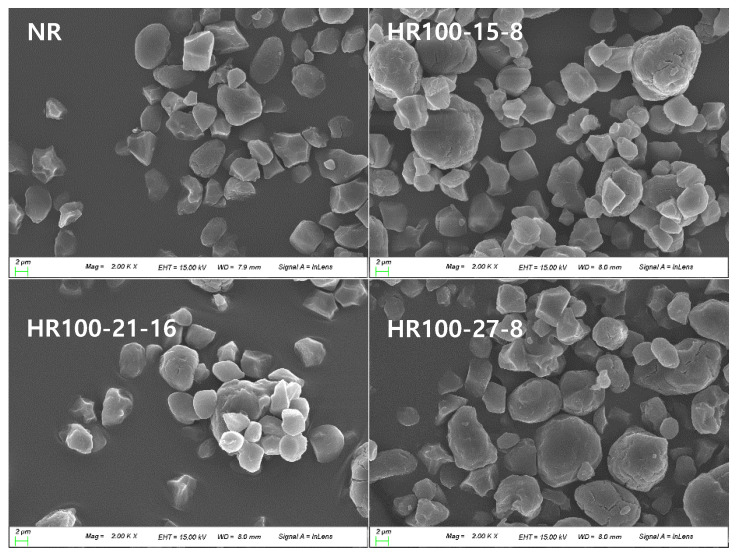
Scanning electron microphotographs of native and heat–moisture treated rice starches. NR: native rice starch, HR: heat–moisture treated rice starch, 100, 110 and 120: heat–moisture treatment temperature, 15, 21 and 27: moisture content, 8 and 16: heat–moisture treatment time.

**Figure 3 molecules-27-06375-f003:**
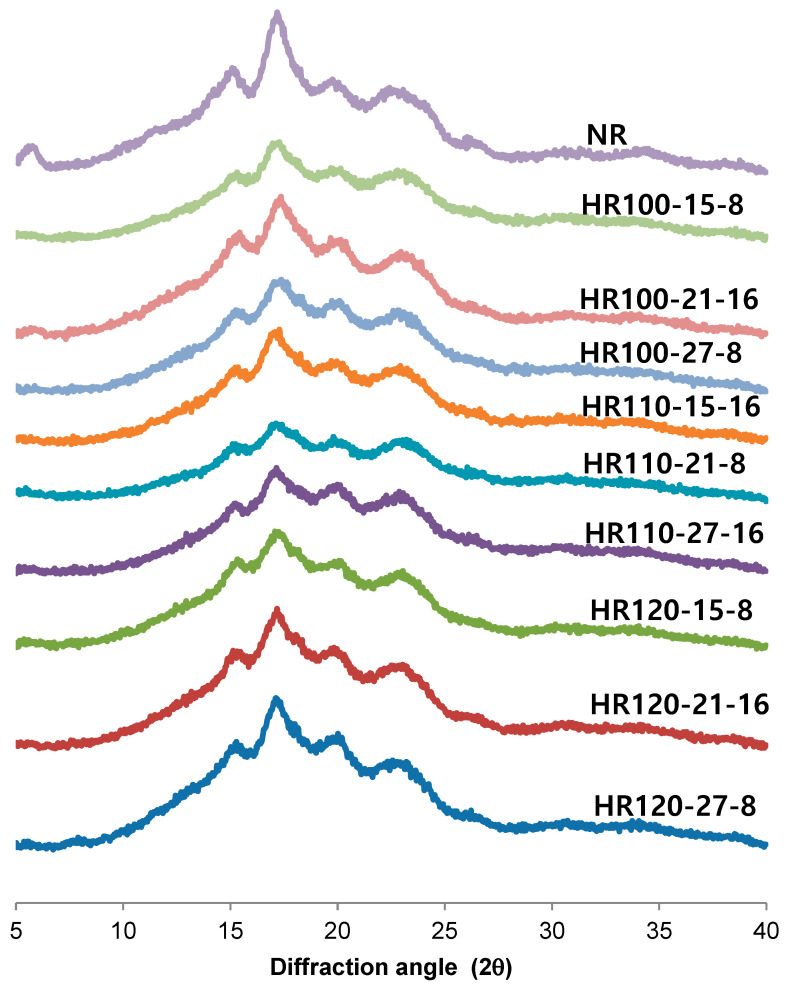
XRD patterns of native and heat–moisture treated rice starches. NR: native rice starch, HR: heat–moisture treated rice starch, 100, 110 and 120: heat–moisture treatment temperature, 15, 21 and 27: moisture content, 8 and 16: heat–moisture treatment time.

**Table 1 molecules-27-06375-t001:** RS content, solubilities and swelling powers of heat–moisture treated rice starch under different treatment conditions.

Starch ^1^	RS Content (%)	Solubility at 80 °C (%)	Swelling Power at 80 °C (g/g)
NR	32.13 ± 0.49 ^d^	6.18 ± 1.33 ^ab^	8.48 ± 0.12 ^a^
HR100-15-8	24.84 ± 0.64 ^f^	4.66 ± 0.14 ^cd^	6.17 ± 0.05 ^bc^
HR100-21-16	49.00 ± 0.12 ^a^	6.70 ± 0.93 ^a^	6.56 ± 0.07 ^b^
HR100-27-8	41.18 ± 0.04 ^b^	3.94 ± 0.08 ^de^	5.66 ± 0.06 ^d^
HR110-15-16	26.58 ± 1.06 ^e^	4.64 ± 0.17 ^cd^	5.79 ± 0.01 ^cd^
HR110-21-8	41.75 ± 0.42 ^b^	4.08 ± 0.34 ^cde^	5.40 ± 0.11 ^de^
HR110-27-16	31.17 ± 0.79 ^d^	4.36 ± 0.62 ^cde^	5.44 ± 0.15 ^de^
HR120-15-8	35.22 ± 0.38 ^c^	5.36 ± 0.28 ^bc^	6.36 ± 0.47 ^b^
HR120-21-16	35.43 ± 1.74 ^c^	3.18 ± 0.08 ^e^	5.10 ± 0.15 ^ef^
HR120-27-8	30.52 ± 0.85 ^d^	3.56 ± 0.00 ^de^	4.91 ± 0.09 ^f^

^1^ RS: resistant starch, NR: native rice starch, HR: heat–moisture treated rice starch, 100, 110 and 120: heat–moisture treatment temperature, 15, 21 and 27: moisture content, 8 and 16: heat–moisture treatment time. ^a–f^ Mean ± SD with different superscript in the same column indicate significant differences (*p* < 0.05).

**Table 2 molecules-27-06375-t002:** Thermal properties of native and heat–moisture treated rice starches under different treatment conditions.

Starch ^1^	*T_o_* (°C)	*T_p_* (°C)	*T_c_* (°C)	*T_c_−T_o_* (°C)	*ΔH* (J/g)
NR	59.6 ± 0.6 ^e^	66.5 ± 0.3 ^i^	74.1 ± 1.1 ^e^	14.6 ± 0.5 ^d^	12.5 ± 1.5 ^a^
HR100-15-8	76.3 ± 0.1 ^d^	83.5 ± 0.0 ^g^	91.6 ± 0.5 ^cd^	15.3 ± 0.6 ^d^	4.2 ± 0.0 ^f^
HR100-21-16	75.6 ± 0.3 ^d^	82.3 ± 0.1 ^h^	89.6 ± 0.4 ^d^	14.0 ± 0.1 ^d^	5.2 ± 0.6 ^ef^
HR100-27-8	80.6 ± 2.7 ^bc^	95.2 ± 1.3 ^a^	107.0 ± 0.1 ^a^	26.4 ± 2.6 ^abc^	8.9 ± 0.6 ^b^
HR110-15-16	78.2 ± 0.1 ^cd^	90.6 ± 0.4 ^d^	106.6 ± 2.1 ^a^	28.5 ± 2.2 ^ab^	8.3 ± 0.0 ^bc^
HR110-21-8	81.3 ± 0.3 ^b^	88.7 ± 0.1 ^e^	95.7 ± 0.8 ^c^	14.4 ± 0.6 ^d^	6.0 ± 0.1 ^de^
HR110-27-16	80.5 ± 2.3 ^bc^	87.0 ± 0.1 ^f^	106.1 ± 0.6 ^a^	25.6 ± 1.7 ^bc^	6.8 ± 0.4 ^cd^
HR120-15-8	77.1 ± 0.4 ^d^	93.7 ± 0.8 ^b^	108.0 ± 4.6 ^a^	30.9 ± 4.2 ^a^	8.4 ± 0.2 ^bc^
HR120-21-16	82.9 ± 0.2 ^ab^	92.2 ± 0.1 ^c^	105.1 ± 3.7 ^a^	22.3 ± 3.9 ^c^	8.3 ± 0.1 ^bc^
HR120-27-8	84.9 ± 1.2 ^a^	92.2 ± 0.4 ^c^	100.4 ± 0.6 ^b^	15.5 ± 0.6 ^d^	9.9 ± 0.6 ^b^

^1^ NR: native rice starch, HR: heat–moisture treated rice starch, 100, 110 and 120: heat–moisture treatment temperature, 15, 21 and 27: moisture content, 8 and 16: heat–moisture treatment time. ^a–i^ Mean ± SD with different superscript in the same column indicate significant differences (*p* < 0.05).

## Data Availability

The data presented in this study are available on request from the corresponding author. The data are not publicly available due to ethical restrictions and intellectual property issues.

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
