# Peer review of "Enhancing Resistant Starch Content of High Amylose Rice Starch through Heat–Moisture Treatment for Industrial Application"

_molecules, 2022, doi:10.3390/molecules27196375_

Round 1

Reviewer 1 Report (Previous Reviewer 2)

The authors answered my questions and improved the quality of the manuscript.

Author Response

Reviewer #1: The authors answered my questions and improved the quality of the manuscript.

Response: Thanks for your positive comment for this study.

Reviewer 2 Report (New Reviewer)

I recommend authors to use “rice starch” instead of “goami starch” if possible, as it is easier to understand the experiment without having to look in material section for “goami” meaning. More, there are places in the manuscript were authors used “rice starch”.

Lines 61-62: „The RS is defined as starch and its derivatives that are undigested in the digestive system of healthy individuals.” The sentence can`t be understood, it seems that the idea was not finished.

Table 1. Swelling power values are not expressed in g/g, as stated in table header.

Line 174. “…resulting it more ordered double helical amylopectin side chain clusters”. It should be “…resulting in more…”

Author Response

We would like to express our appreciation to Editor and the Reviewers for the comments and suggestions how to improve the quality of the manuscript. A detailed list of queries and answers to the specific issues addressed by the reviewers follows. Changes and amendments according to the reviewers’ guidelines have been marked by red in the revised manuscript.

Reviewer #1 : I recommend authors to use “rice starch” instead of “goami starch” if possible, as it is easier to understand the experiment without having to look in material section for “goami” meaning. More, there are places in the manuscript were authors used “rice starch”.

Response: Thanks for your valuable comment. Definitely, we agree with your precise comment, thereby “goami starch” was changed to “rice starch” through an entire manuscript.

Lines 61-62: „The RS is defined as starch and its derivatives that are undigested in the digestive system of healthy individuals.” The sentence can`t be understood, it seems that the idea was not finished.

Response: Thanks for your valuable comment. This sentence was revised as follows.

[L60-63, Starch in foods can be classified as rapidly digestible starch, slowly digestible starch, and resistant starch (RS) according to its degree and speed of digestion. The RS is a fraction of the starch that is not digested in the small intestine of healthy individuals and enters the colon unchanged, where fermentation by resident microorganisms occurs, resulting in the formation of short-chain fatty acids [7].]

Table 1. Swelling power values are not expressed in g/g, as stated in table header.

Response: Thanks for your precise comment. In L337, the unit for swelling power was expressed as g/g.

Line 174. “…resulting it more ordered double helical amylopectin side chain clusters”. It should be “…resulting in more…”

Response: Apologies for this mistake. The word was correctly revised.

This manuscript is a resubmission of an earlier submission. The following is a list of the peer review reports and author responses from that submission.

Round 1

Reviewer 1 Report

This work present Optimization of Heat Moisture Treatment for Improving the Resistant Starch Content of High Amylose Rice Starch.  This manuscript need changes which are mentioned as following

Paper format: The paper is in Molecules format.

Bytheway I would to like to submit such work in MDPI Foods. Anyhow Ok here

Title: The MS title sound good, however, need to change to change the title. Indicate the objective of the work in title. Change please

Abstract: The abstract should always be concise and informative. The arguments of why your study is important not making any sense. Extensive revision is required in abstract, as the present sentences sounds noisy during reading. There are grammatical errors and need to minimize the sentences in length. Overall the abstract is not informative enough and need to show the actual picture of the work. Authors are supposed to please indicate the numerical values with significant difference in abstract.

Keywords: Need to change the keywords, these keywords are not enough for researching in search engine.

Introduction: Introduction is supported with nice arguments, however some shortcoming are below

1.     Resistance starch should define first, why RS were considered in this study? There are soluble starch and insoluble starch, what about these types?

2.     Up to my knowledge there are four different types of RS, I haven’t got any information which type RS was targeted in this study

3.     I don’t get any information why such work have benefits over other methods?

4.     Is Heat treatment change the RS behavior? Means RS will release in glucose molecules slowly? Nothing such information has been provided.

5.     What is the safety and environment concern for heat treatment of starch

6.     Please add/indicate and compare your work with pervious published paper.

7.     Please cite the following latest paper

https://link.springer.com/article/10.1007/s10924-020-01837-1

Methods and materials:

1.     Why quadratic model was used to determine the different values? It always makes result difficult to understand, one can go through table to define these values.

2.     Lines 260-267 needs revision, put details of the experiment.

3.     No equation has been provided for swelling and solubility experiments.

4.     Why only RS was calculated? What about amylose contents in sample? Why it was ignore to count?

5.     No experiments were conducted for glycemic index (GI) of RS, as such calculation always done for low GI values starch.

Results and discussion: Results are fine but first need the experiment which are suggested above Conclusion: Change the conclusion as per changes in discussion, add more of application for this research for common reader.

Author Response

Editors comments

We would like to express our appreciation to Editor and the Reviewers for the comments and suggestions how to improve the quality of the manuscript. A detailed list of queries and answers to the specific issues addressed by the reviewers follows. Changes and amendments according to the reviewers’ guidelines have been marked by red in the revised manuscript.

Reviewers' comments:

Reviewer #1: This work present Optimization of Heat Moisture Treatment for Improving the Resistant Starch Content of High Amylose Rice Starch.  This manuscript need changes which are mentioned as following

  1. Paper format: The paper is in Molecules format.By the way I would to like to submit such work in MDPI Foods. Anyhow Ok here.

Response: Thanks for your comment. This article was supposed to be written for submission in special issue “Starch chemistry in Food Products” of Molecules.

  1. Title: The MS title sound good, however, need to change to change the title. Indicate the objective of the work in title. Change please.

Response: Thanks for your useful comment. The title was revised to clarify the purpose of this study. The purpose of this study was to establish the conditions for hydrothermal treatment of high amylose rice starch with increased resistant starch content to improve the industrial applicability of domestic high amylose rice.

[Enhancing Resistant Starch Content of High Amylose Rice Starch through Heat-Moisture Treatment for Industrial Application]

  1. Abstract: The abstract should always be concise and informative. The arguments of why your study is important not making any sense. Extensive revision is required in abstract, as the present sentences sounds noisy during reading. There are grammatical errors and need to minimize the sentences in length. Overall the abstract is not informative enough and need to show the actual picture of the work. Authors are supposed to please indicate the numerical values with significant difference in abstract.

Response: Definitely, we agree with your suggestion. The abstract was extensively revised by considering your

advice.

[The objective of the present study is to enhance the resistant starch (RS) content of high amylose rice starch with heat-moisture treatment (HMT) for industrial application. The optimized HMT condition for achieving the highest RS content (46.4%) established using response surface methodology (RSM) was a temperature of 100 oC, moisture content of 24.2%, and time of 11.5 h. Upon HMT, the RS content increased from 32.1% for native starch to 46.4% in HMT starch with optimized condition. HMT of the starches reduced the solubility and swelling power. The surface of HMT starch granules was more irregular than native starch. The X-ray diffraction (XRD) peak intensity at 2θ = 5° was greatly reduced by HMT, and the peaks at 22.7° and 24.2° were merged. HMT increased the gelatinization temperature and reduced the gelatinization enthalpy. HMT provide a method for production of high-yield RS2 with high amylose rice starch in indus-trial application.]

  1. Keywords: Need to change the keywords, these keywords are not enough for researching in search engine.

Response: We thank reviewer comment. We have revised keywords by fully following your comments as follows.

[resistant starch; heat-moisture treatment; high amylose rice starch; response surface methodology; physicochemical properties; granular morphology]

  1. Introduction: Introduction is supported with nice arguments, however some shortcoming are below.

Response: Thanks for your useful comment and concern for this manuscript. We have revised the introduction section by fully following your comments.

5.1. Resistance starch should define first, why RS were considered in this study? There are soluble starch and insoluble starch, what about these types?

Response: Thanks for your useful comment and concern for this study. We have revised manuscript by fully following this comment.

[Regarding its degree and speed of digestion, starch can be divided into rapidly digestible starch, slowly digestible starch, and resistant starch (RS). The RS is defined as starch and its derivatives that are undigested in the digestive system of healthy individuals. RS is classified into subgroups, RS1RS5. RS1 represents native starch that is in a physically inaccessible with enzymes. RS2 represents potato starch and high amylose starches with B-type structure that are resistant to enzyme digestion. RS3 is retrograded starch during cooling of gelatinized starch. RS4 is chemically modified starch by various types of chemical treatments. RS5 is a type of RS arising from the formation of amylose-lipid complexes [7]. The high-RS foods have gained considerable interest by consumers because its consumption could effectively prevent obesity, diabetes, cardiovascular dis-ease and other chronic diseases [7]. A lot of insoluble RS4 type starches have been developed as a food source of high-RS foods. High amylose starch can be classified as RS2 due to its high enzyme resistance. As the amylose content increased, the susceptibility to enzymatic hydrolysis was reduced. The high-yield RS prepared by heat-moisture treatment and multiple heating and cooling cycles with high amylose corn starch have been developed and marketed [8-11]. This process could be suitable to production of industrial-scale RS2 type starch at low cost.]

5.2. Up to my knowledge there are four different types of RS, I haven’t got any information which type RS was targeted in this study.

Response: Thanks for your valuable comment. The objective of this study was to establish the optimal conditions for hydrothermal treatment of high amylose rice starch (RS type 2) with enhancing RS content in order to improve the industrial applicability of domestic high amylose rice. The following content was suggested in revised manuscript.

[The high-yield RS prepared by heat-moisture treatment and multiple heating and cooling cycles with high amylose corn starch have been developed and marketed [8-11]. This process could be suitable to production of industrial-scale RS2 type starch at low cost. The objective of this study was to establish the optimal conditions for hydrothermal treatment of high amylose rice starch with enhancing RS content in order to improve the industrial applicability of domestic high amylose rice.]

5.3. I don’t get any information why such work have benefits over other methods?

Response: Thanks for your useful comment. HMT is a physical modification that would be suitable industrial application. The related explanation was added in introduction section.

[HMT is relatively easy to implement industrial production due to its simple process and ease of management. Whereas, annealing, hydrothermal treatment of starch in the presence of excess water for an extend period of time, has a high risk of decomposition and multiple heating and cooling cycles, which are difficult to be industrialized due to complicated processes and high utility costs.]

[Most studies on high amylose rice starch have involved the structural and physicochemical properties of rice starch [10, 11] and effects of heat-moisture treatment on functional properties of rice starch [4, 12]. To date, no studies have explored the suitable meth-od of RS-rich preparation with high amylose rice starch for industrial application. The objective of this study was to establish the optimal conditions for hydrothermal treatment of high amylose rice starch with enhancing RS content in order to improve the industrial applicability of domestic high amylose rice. Furthermore, the physicochemical properties of heat-moisture treated high amylose rice starches was investigated.]

5.4. Is Heat treatment change the RS behavior? Means RS will release in glucose molecules slowly? Nothing such information has been provided.

Response: Thanks for your valuable comment. In introduction section, the effect of HMT on RS was introduced.

[High amylose starch can be classified as RS2 due to its high enzyme resistance. As the amylose content increased, the susceptibility to enzymatic hydrolysis was reduced. The high-yield RS prepared by heat-moisture treatment and multiple heating and cooling cycles with high amylose corn starch have been developed and marketed [8-11]. This process could be suitable to production of industrial-scale RS2 type starch at low cost.]

[To date, no studies have explored the suitable method of RS-rich preparation with high amylose rice starch for industrial application. The objective of this study was to establish the optimal conditions for hydrothermal treatment of high amylose rice starch with enhancing RS content in order to improve the industrial applicability of domestic high amylose rice.]

[HMT starches improve heat and shear stability characteristic like cross-linked starches. These physicochemical changes imply that HMT starch could be used as resistant starch (RS) resources.]

5.5. What is the safety and environment concern for heat treatment of starch.

Response: Thanks for your valuable comment. The explanation for this comment was added in revised manuscript as follows.

[However, chemical modification has become less favorable among consumers because by-products of chemical reagents are present in the modified starch. By contrast the physical treatments of starch have been gaining increased acceptance. The primary interest in physical modification is related to the clean label starches; they can be classified as natural products even though they are processed [2]. In addition, physical modification in starch has advantages of its effectiveness, flexibility in relation to heat sources, low cost and the non-generation of chemical residues, which make it extremely attractive method for industries.]

5.6. Please add/indicate and compare your work with pervious published paper.

Response: Thanks for your useful comment. We agree with your suggestion. Based on this comment, we revised the manuscript as follows.

[A lot of insoluble RS4 type starches have been developed as a food source of high-RS foods. High amylose starch can be classified as RS2 due to its high enzyme resistance. As the amylose content increased, the susceptibility to enzymatic hydrolysis was reduced. The high-yield RS prepared by heat-moisture treatment and multiple heating and cooling cycles with high amylose corn starch have been developed and marketed [8-11]. This process could be suitable to production of industrial-scale RS2 type starch at low cost.

Most studies on high amylose rice starch have involved the structural and physicochemical properties of rice starch [10, 11] and effects of heat-moisture treatment on functional properties of rice starch [4, 12]. To date, no studies have explored the suitable meth-od of RS-rich preparation with high amylose rice starch for industrial application.]

5.7. Please cite the following latest paper

https://link.springer.com/article/10.1007/s10924-020-01837-1

Response: Thanks for your recommendation in above paper. We have read the paper “Green Production and Structural Evaluation of Maize Starch–Fatty Acid Complexes Through High Speed Homogenization”. Unfortunately, we could not find the contents that could be cited in our manuscript.

  1. Methods and materials:

6.1. Why quadratic model was used to determine the different values? It always makes result difficult to understand, one can go through table to define these values.

Response: Thanks for your useful comment and concern for this manuscript. Box-Behnken design was employed to understand the effects of temperature, moisture content, and time on the yield of RS. The advantage of this model is that the experiment can be designed within the scope initially set by the designer. This can increase the efficiency of research by reducing the risk of experiments that cannot be implemented in the industrial field. The ANOVA of the independent variables for the quadratic model was add on supplementary table.

6.2. Lines 260-267 needs revision, put details of the experiment.

Response: Thanks for your useful comment and concern for this manuscript. The preparation of HMT starch samples was totally revised to describe in more detail.

[After measuring the moisture content of raw rice starch, starch samples (20 g) were weighed into the screw cap glass container (100 ml). The starch was adjusted to 15, 21, and 27% moisture content by adding the appropriate amount of distilled water. The samples were mixed thoroughly to breakdown clumps. The containers were sealed, kept at ambient temperature for 24 h to equilibrate the moisture in starch, then placed in a forced air oven (VS-1202D3, Vision Scientific Co., Bucheon, Korea) at 100, 110, and 120°C. The samples were thermally treated for 8 and 16 h at each temperature. Afterwards the containers were opened, and the starch samples were air-dried until moisture content reached around 10%. The samples were ground and passed through a 120 mesh (<125 mm) sieve, and stored in a desiccator until they were used.]

6.3. No equation has been provided for swelling and solubility experiments.

Response: Thanks for your comment. The equations were added in the revised manuscript.

6.4. Why only RS was calculated? What about amylose contents in sample? Why it was ignore to count?

Response: Thanks for your valuable comments. Definitely, apparent amylose contents of rice starches are an influencing factor on RS content in starch. The previous study also reported a decrease in amylose content of starch after HMT. However, the objective of this study was to establish the optimal conditions for hydrothermal treatment of high amylose rice starch with enhancing RS content in order to improve the industrial applicability of domestic high amylose rice. The enhancing RS content upon HMT could be mainly attributed to crystalline perfection. For this reason, the determined physicochemical properties in this study were solubility, swelling factor, granular morphology, XRD crystalline structure, thermal properties for the analysis of crystalline structure during HMT.

6.5. No experiments were conducted for glycemic index (GI) of RS, as such calculation always done for low GI values starch.

Response: Definitely, we agree with your suggestion. The enhanced RS content by HMT could be linked to the decreased GI value. the objective of this study was to establish the optimal conditions for hydrothermal treatment of high amylose rice starch with enhancing RS content in order to improve the industrial applicability of domestic high amylose rice using RSM. For this reason, in our study, RS level of high amylose rice starch was only used as the dependent variable.

  1. Results and discussion: Results are fine but first need the experiment which are suggested above Conclusion: Change the conclusion as per changes in discussion, add more of application for this research for common reader.

Response: We are grateful for the reviewer’s constructive and useful comment. The format of “Molecules” journal has suggested the order in “Results and discussion”, “Materials and methods”, and “Conclusions”.

 We have extensively revised the section of “Conclusions” by fully following your comment.

[To improve the industrial applicability of high amylose rice starch, RS content was increased through HMT and the physicochemical properties of HMT starches were investigated. This study clearly showed that optimal HMT condition for enhancing RS content was established using RSM. HMT leads to an increase in crystalline perfection due to strengthening the bonds between amylose molecules and between amylose and amylopectin of starch granules because the solubility and swelling power of the starch granules decreased, and the gelatinization temperature increased during HMT.  Our results could provide useful information for establishing the HMT conditions of high amylose rice starch for the production of high-RS food in industrial application of rice materials.]

Reviewer 2 Report

In this study, the authors determined the optimal HMT condition for improving the RS content of high amylose rice starch, and also investigated the properties of treated starch. However, the authors failed to clearly explain the purpose and the significance of this study as HMT has been a very common method for improving RS content. Moreover, there are some errors in the data in the article. Therefore, the manuscript should be rejected.

Here are some suggestions and comments for the manuscript:

1.       In the Abstract, the final sentence is important and used for indicating the significance of this study. However, the authors mentioned that “HMT could significantly increase the RS levels of high amylose rice starch”, which has been proved by many other studies.

2.       Line 71: reaction time: 8 and 12 h? Should be 8 and 16 h.

3.       Line 73: The RS content of native rice starch was 32.1%? I have serious doubts about this data. With such high RS content, this native rice starch can be directly applied as a functional starch without any further modifications.

4.       I also have doubts about that HMT could decrease RS content.

Author Response

Reviewer #2: In this study, the authors determined the optimal HMT condition for improving the RS content of high amylose rice starch, and also investigated the properties of treated starch. However, the authors failed to clearly explain the purpose and the significance of this study as HMT has been a very common method for improving RS content. Moreover, there are some errors in the data in the article. Therefore, the manuscript should be rejected. Here are some suggestions and comments for the manuscript:

Response: We thank the reviewer for valuable comment and concern for this manuscript. As suggested, the HMT method is common method for improving RS content for several starch sources. The previous version of our manuscript did not clearly show novelty and significance of our study. We have revised manuscript by considering these points.

Novelty and significant of our study are related as following suggestions.

  1. To date, no studies have explored the suitable method of RS-rich preparation with high amylose rice starch although most studies on high amylose rice starch have involved the structural and physicochemical properties of rice starch and effects of heat-moisture treatment on functional properties of rice starch.
  2. RS-rich preparation with high amylose rice starch (type 2 RS) by using HMT have not been studied.
  3. To our knowledge, our report is the first trial to assess the optimal conditions for enhancing RS content using RSM in order to improve the industrial applicability of domestic high amylose rice.

  1. In the Abstract, the final sentence is important and used for indicating the significance of this study. However, the authors mentioned that “HMT could significantly increase the RS levels of high amylose rice starch”, which has been proved by many other studies.

Response: We thank your valuable comment and concern for this manuscript. We agree with your suggestion. The abstract was extensively revised by considering your advice.

[The objective of the present study is to enhance the resistant starch (RS) content of high amylose rice starch with heat-moisture treatment (HMT) for industrial application. The optimized HMT condition for achieving the highest RS content established using response surface methodology (RSM) was a temperature of 100 oC, moisture content of 24.2%, and time of 11.5 h. Upon HMT, the RS content increased from 32.1% for native starch to 46.4% in HMT starch with optimized condition. HMT of the starches reduced the solubility and swelling power. The surface of HMT starch granules was more irregular than native starch. The X-ray diffraction (XRD) peak intensity at 2θ = 5° was greatly reduced by HMT, and the peaks at 22.7° and 24.2° were merged. HMT increased the gelatinization temperature and reduced the gelatinization enthalpy. HMT provide a method for production of high-yield RS2 with high amylose rice starch in industrial application.]

  1. Line 71: reaction time: 8 and 12 h? Should be 8 and 16 h.

Response: We are deeply appreciated your precise comment. The data was amended.

  1. Line 73: The RS content of native rice starch was 32.1%? I have serious doubts about this data. With such high RS content, this native rice starch can be directly applied as a functional starch without any further modifications.

Response: We are deeply appreciated your valuable comment and concern. Definitely, we respect your doubt on RS content since RS contents of rice starch (~20% amylose content) in our previous studies were below 10%. However, in this study, we used the high amylose rice starch with 32.9% apparent amylose content. Although the RS content of rice starch could be highly influenced the employed method, enzyme concentration, and rice cultivar, the high RS content with high amylose rice starch have been recently reported as follows.

  1. Chen et al. (2022). Foods 11, 94. 36~41% RS content for high amylose rice (26.8~27.7% amylose content)
  2. Liu et al. (2021). Food Hydrocolloids, 113, 106441. 36.8~54.5% RS content of high amylose rice starch (19.3~46.7% amylose content).
  3. Par et al. (2020). Food Hydrocolloids, 102, 105544. 18.22% RS content of high amylose rice starch (51.7% amylose content).

  1. I also have doubts about that HMT could decrease RS content.

Response: Thanks for your valuable comment. The explanation of the decreased RS on HMT was added in revised manuscript. Normally, the increased RS content during HMT have been reported as found in rice starch treated some HMT condition in this study. However, some studies reported reduction in RS after HMT as reported in review paper by Fonseca et al (2021).

[In contrast, the decreased RS on HMT could be attributed to the partial disruption of crystalline structure with evidence of decreased gelatinization enthalpy [6].]  

Round 2

Reviewer 1 Report

Actually the authors were asked to include the GI values. No one can just simply believe with increase in RS value with thermal treatment can be industrial importance. I strongly believe such data must include in this study so can listen the music of the manuscript. I again insist to add GI values and count amylose contents of the sample which will help to clear the picture.

Reviewer 2 Report

The authors answered my questions and improved the quality of this manuscript. Therefore, the manuscript can be accepted after minor revisions.

1.      Line 46-50, the authors should provide reference(s) for supporting the reasons you choose HMT as a relatively easy method in this study.

2.      Line 84: should be changed to “were investigated”.

3.      Could the authors explain why the apparent viscosity of the different starch samples was not determined?